# Genetic diversity of *Trypanosoma cruzi* parasites infecting dogs in southern Louisiana sheds light on parasite transmission cycles and serological diagnostic performance

Eric Dumonteil[1,2]*, Ardem Elmayan[1,2], Alicia Majeau[1,2], Weihong Tu[1,2], Brandy Duhon[3], Preston Marx[1,4], Wendy Wolfson[3], Garry Balsamo[5], Claudia Herrera[1,2]

1 Department of Tropical Medicine, School of Public Health and Tropical Medicine, Tulane University, New Orleans, Louisiana, United States of America, 2 Vector-Borne and Infectious Disease Research Center, Tulane University, New Orleans, Louisiana, United States of America, 3 School of Veterinary Medicine, Louisiana State University, Baton Rouge, Louisiana, United States of America, 4 Division of Microbiology, Tulane National Primate Research Center, Tulane University, Covington, Louisiana, United States of America, 5 Infectious Disease Epidemiology Section, Office of Public Health, Department of Health, New Orleans, Louisiana, United States of America

* edumonte@tulane.edu

## Abstract

### Background

Chagas disease is a neglected zoonosis of growing concern in the southern US, caused by the parasite *Trypanosoma cruzi*. We genotyped parasites in a large cohort of PCR positive dogs to shed light on parasite transmission cycles and assess potential relationships between parasite diversity and serological test performance.

### Methodology/principal findings

We used a metabarcoding approach based on deep sequencing of *T. cruzi* mini-exon marker to assess parasite diversity. Phylogenetic analysis of 178 sequences from 40 dogs confirmed the presence of *T. cruzi* discrete typing unit (DTU) TcI and TcIV, as well as TcII, TcV and TcVI for the first time in US dogs. Infections with multiple DTUs occurred in 38% of the dogs. These data indicate a greater genetic diversity of *T. cruzi* than previously detected in the US. Comparison of *T. cruzi* sequence diversity indicated that highly similar *T. cruzi* strains from these DTUs circulate in hosts and vectors in Louisiana, indicating that they are involved in a shared *T. cruzi* parasite transmission cycle. However, TcIV and TcV were sampled more frequently in vectors, while TcII and TcVI were sampled more frequently in dogs.

### Conclusions/significance

These observations point to ecological host-fitting being a dominant mechanism involved in the diversification of *T. cruzi*-host associations. Dogs with negative, discordant or confirmed positive *T. cruzi* serology harbored TcI parasites with different mini-exon sequences, which strongly supports the hypothesis that parasite genetic diversity is a key factor affecting

**Data Availability Statement:** Mini-exon sequences from dogs have been deposited in the GenBank database (Accession # MT365251-MT365424).

**Funding:** This work was supported by a COBRE Pilot grant from the LSU-Tulane COBRE Center for Experimental Infectious Disease Research program, grant #632083 from Tulane University School of Public Health and Tropical Medicine, and the Louisiana Board of Regents through the Board of Regents Support Fund [# LESASF (2018-21)-RD-A-19] to E.D. We also acknowledge the National Center for Research Resources and the Office of Research Infrastructure Programs (ORIP) of the NIH through grant P51 OD011104 to the Tulane National Primate Research Center. The funders had no role in study design, data collection and analysis, decision to publish, or preparation of the manuscript.

**Competing interests:** The authors have declared that no competing interests exist.

serological test performance. Thus, the identification of conserved parasite antigens should be a high priority for the improvement of current serological tests.

## Author summary

Chagas disease is a neglected zoonosis of growing concern in the southern US, caused by the parasite *Trypanosoma cruzi*. Here we analyzed the parasite genetic diversity in a large cohort of infected dogs to better understand parasite transmission cycles and assess potential relationships between parasite diversity and serological test performance. We used DNA sequencing of a well characterized *T. cruzi* genetic marker to assess parasite diversity. We confirmed the presence of *T. cruzi* lineages TcI and TcIV, and report TcII, TcV and TcVI for the first time in US dogs. Parasite lineages TcIV TcII and TcVI appeared more frequent in dogs compared to insect vectors. Dogs with negative, discordant or confirmed positive *T. cruzi* serology harbored genetically different TcI parasites, which shows that parasite genetic diversity is a key factor affecting serological test performance. Thus, the identification of parasite antigens conserved across strains and lineages should be a high priority for the improvement of current serological tests.

## Introduction

Chagas disease is a neglected zoonosis caused by the protozoan parasite *Trypanosoma cruzi*, which is endemic in the Americas. The parasite is transmitted to mammalian hosts by the infected feces of triatomine bugs, also known as kissing bugs. Infection leads to chronic cardiomyopathy in 20–40% of humans. Dogs have been identified as a major domestic/peridomestic host for *T. cruzi* infection in many countries and epidemiological setting. They can play an important role as a domestic reservoir of the parasite, and can contribute to an increased risk for *T. cruzi* infection in humans [1–4]. Dogs can also develop severe chronic cardiomyopathy with poor prognosis [5,6].

In the US, zoonotic cycles are well established in the southern half of the country, and the parasite can be transmitted by multiple species of triatomines, with *Triatoma gerstaeckeri* and *Triatoma sanguisuga* as some of the most common vector species [7]. *Trypanosoma cruzi* infection in dogs has been particularly well studied in Texas, with an average seroprevalence of infection around 8%, but varying greatly depending on the dog population studied [8]. In Louisiana, we recently reported a seroprevalence of *T. cruzi* antibodies of 7% in shelter dogs [9], in agreement with the widespread circulation of *T. cruzi* in this region. In spite of this high level of infection in dogs, there is still limited awareness of the disease among health professionals, and as a consequence, only a few canine and human cases are sporadically detected [10].

A major limitation for improved surveillance of *T. cruzi* infection in dogs is the lack of reliable serological tests, as significant discordance among tests is frequently observed. For example, in a serological survey of trained government dogs from the Department of Homeland Security in southern Texas, the seroprevalence of infection varied from 7.4 to 18.9%, depending on the tests used [11]. Similarly in Louisiana, we found that only 7% of dogs PCR positive for *T. cruzi* are also confirmed positive by serology (at least two reactive tests), while 41% have discordant serological test results (only one reactive test) and 52% are completely seronegative,

based on three serological tests [9]. Discordance among tests has also been observed for human samples [12–14] and it is becoming a key issue for the reliable surveillance of Chagas disease and patient care in North America [15,16].

Some of this discordance has been attributed to mismatches between antigens used for serological diagnostic and the parasite strains actually circulating in mammalian hosts. Indeed, *T. cruzi* presents a wide genetic diversity, which has led to its division into seven discrete typing units (DTUs) referred to as TcI to TcVI and TcBat [17]. While initial studies only uncovered the presence of TcI and TcIV in the US [18], improvement in genotyping methods and the development of next-generation sequencing and metabarcoding approaches [19–21] have led to the identification of additional DTUs including TcII, TcV and TcVI. These studies suggest that current models of parasite genotype distribution in the country need to be reassessed to allow for a reliable understanding of the risk for *T. cruzi* infection in both dogs and humans [22]. In particular, metabarcoding appeared very powerful to uncover infections with multiple parasite haplotypes that are missed by conventional approaches [23].

Therefore, our objective was to identify *T. cruzi* genetic diversity circulating in shelter dogs from southern Louisiana, to shed light on parasite transmission cycles in this region. We also assessed potential relationships between parasite genetic diversity and serological test performance, to better understand tests shortcomings.

## Methods

### Ethics statement

The study received approval from Louisiana State University and Tulane University Institutional Animal Care and Use Committees (IACUC).

### Dog blood sample collection and analysis

Archived blood DNA samples from a previous study on the prevalence of *T. cruzi* infection in shelter dogs were used for *T. cruzi* parasite genotyping [9]. These were derived from a convenience sampling of 540 animals from 20 shelters from southern Louisiana, including from Acadia, Ascension, Calcasieu, East Baton Rouge, Iberia, Iberville, Jackson, Lafourche, Livingston, Natchitoches, Orleans, St. Landry, St. Martin and Tangipahoa parishes (S1 Fig). Purified DNA was stored at -20˚C until used, and a total of 73 samples that were PCR positive for *T. cruzi* were included in this study. Four (5%) of these PCR positive dogs were confirmed seropositive (at least two positive tests including InBios Stat-Pak rapid test, homemade ELISA and western blot), 39 (53%) had discordant serology (a single positive test) and 30 (41%) were seronegative with all tests [9].

### Genotyping and deep sequencing

To minimize the risk of sample contamination, DNA extractions and PCR reactions were performed in dedicated cabinets located in separate laboratory rooms, distinct from where gel electrophoresis was performed. Rigorous internal quality controls (e.g., negative and positive controls) were included in each PCR runs. Samples were genotyped by PCR amplification of the mini-exon sequence using a multiplex PCR as described by Souto et al., which gives PCR products of different sizes according to the DTU [24], as well as with newly designed primers that amplify a larger fragment of 500 bp of this marker from all DTUs [19]. Following end-repair and indexing of the PCR amplicons, libraries were prepared and sequenced on a MiSeq

(Illumina) platform. From 100,000 to 600,000 paired reads were obtained from each dog after quality filtering.

## Sequence and data analysis

Raw Fastq sequences files were imported into Geneious 11 software for analysis. Sequences were filtered for quality and length, and reads from each dog were competitively mapped to mini-exon reference sequences from each parasite DTU as previously described [23], to ensure optimum detection of sequences matching the respective DTUs. Reference sequences used were TcI: Raccoon70 (EF576837), TcII: Tu18 (AY367125), TcIII: M5631 (AY367126), TcIV: 92122102r (AY367124), TcV: SC43 (AY367127), TcVI: CL (U57984) and TcBat: TCC2477cl1 (KT305884). Assemblies to each reference DTU were then analyzed separately. Partial matches to a DTU were discarded from the analysis, and only assemblies covering the full-length of the expected mini-exon PCR products were considered to ensure specificity. Mini-exon sequences were then trimmed of PCR primer sequences. Next, sequence variants were identified using FreeBayes SNP/variant tool [25] and only sequences representing at least 1% of the reads were conserved for analysis. Mini-exon sequences from dogs have been deposited in the GenBank database (Accession # MT365251-MT365424, S1 Table). Phylogenetic trees based on maximum likelihood were built using PHYML as implemented in Geneious and mini-exon sequences from reference parasite strains from all DTUs were included for comparison (TcI: Raccoon70 (EF576837), SilvioX10 (CP015667), P/209cl1 (EF576816), TcII: Tu18 (AY367125), TcIII: M5631 (AY367126), TcIV: 92122102r (AY367124), CanIII (AY367123), MT4167 (AF050523), TcV: SC43 (AY367127) and TcVI: CL (U57984). Templeton, Crandall and Sing (TCS) mini-exon haplotype networks [26] were elaborated in PopArt for individual dogs to illustrate haplotype diversity. Parasite mini-exon sequences from dogs were also compared to those from *Triatoma sanguisuga* vectors as well as other vertebrate hosts from southern Louisiana to assess their similarity. Triatomines were from 10 resident houses in southern Louisiana and six animal shelters (two of which were included in this study) [27]. Rodents were from urban and rural New Orleans [21], and non-human primates were from the Tulane National Primate Research Center in Covington, LA [20]. The list of sequences used in the comparisons is indicated in S2 Table. We used a principal component analysis to visualize sequence similarity among hosts and vectors and tested for statistical significance of differences among these by permutational MANOVA (PERMANOVA) based on 10,000 permutations, which does not rely on particular assumptions on data distribution. Analyses were performed using PAST 4.03. We also elaborated TCS haplotype networks to identify haplotypes that may be shared among species. DTU proportions in dog and vectors were also compared by $X^2$ test, and the Berger-Parker dominance index was calculated to assess haplotype distribution. Finally, TcI mini-exon sequences were also compared among groups of dogs with seropositive, seronegative and discordant serological tests. Serological testing was based on three tests: Stat-Pak rapid immunochromatographic test, an in-house ELISA and western blot tests using whole parasite lysate from a local TcI strain (WB1) as antigen [9]. Cut-off values for the ELISA we defined as the mean of negative controls plus three standard deviations. TCS haplotype networks [26] were elaborated in PopArt to assess differences in mini-exon sequences among groups of dogs. The pairwise genetic distance of each mini-exon sequence with that from the WB1 strain was also compared among dogs with seropositive, seronegative and discordant serological tests, to evaluate if increased mini-exon genetic distance was associated with increased discordance and/or seronegative serology. We also used principal component analysis of TcI sequences to compare them among these groups of dogs and statistical significance of the differences was assessed by PERMANOVA as above.

## Results

### *T. cruzi* sequence diversity in dogs

In total, 73 samples from *T. cruzi* PCR positive dogs were included for genotyping using the mini-exon marker, and we were able to recover 178 quality mini-exon sequences from 40 samples (55%). Dogs harbored an average 4.5 ± 0.6 sequence haplotypes/dog, ranging from 1 to 18 sequence haplotypes, at variable frequencies. We performed a phylogenetic analysis of these sequences, together with sequences from reference strains (Fig 1). The initial analysis of all the sequences indicated several well-supported clusters of sequences corresponding to *T. cruzi* parasites DTU TcI, TcII, TcIV, TcV and TcVI (Fig 1A). In addition, some sequence diversity and substructuring could be detected within each DTU for which there were multiple sequences. Separate phylogenetic analyses of DTUs TcI and TcIV confirmed this intra-DTU diversity and the strong divergence between TcIV strains from north and south America (S2 Fig). Detailed analysis of sequences from TcII, TcV and TcVI confirmed that multiple sequences from dog parasites clustered with TcII and TcVI reference sequences, and a single

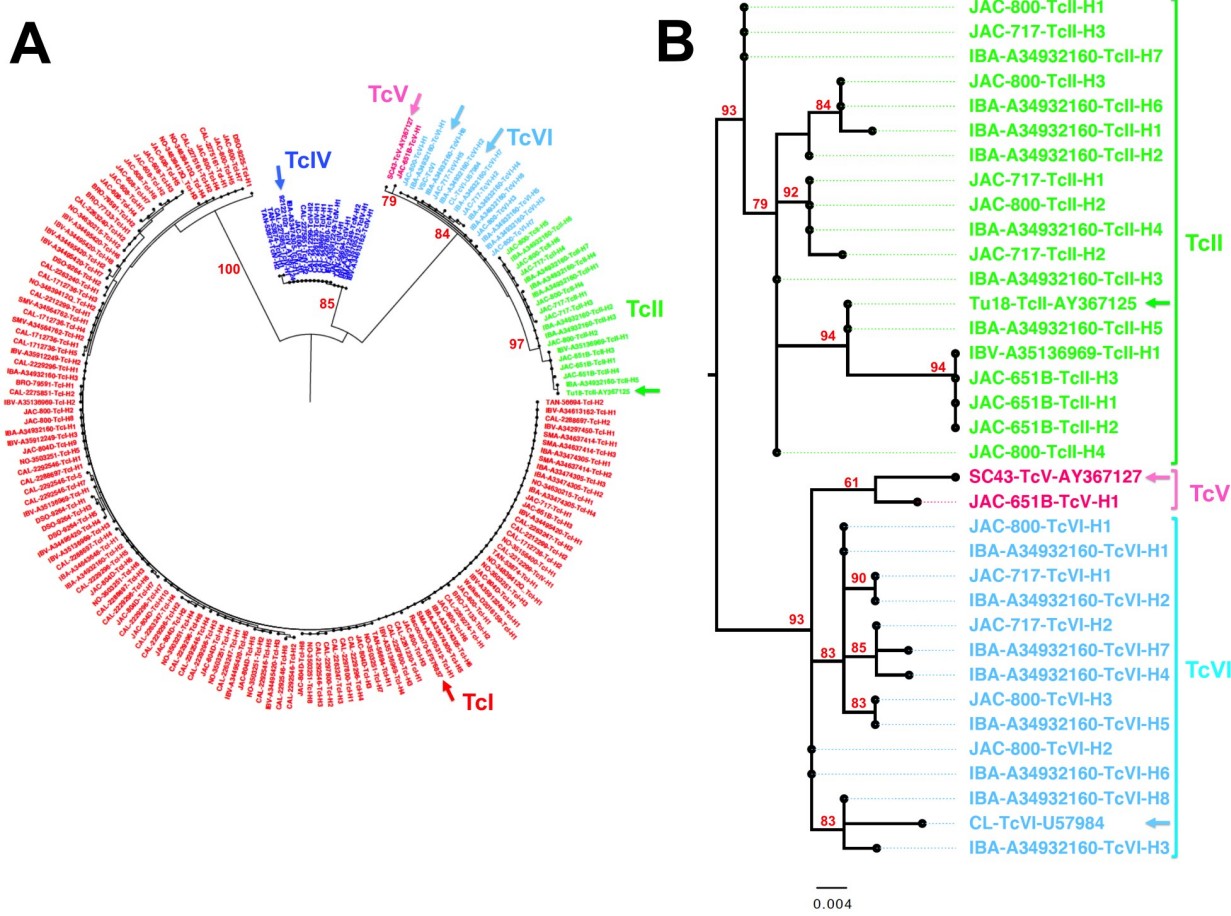

**Fig 1. Phylogenetic analysis of *T. cruzi* sequences.** (**A**) Maximum likelihood analysis of 178 parasite mini-exon sequences derived from shelter dogs resulted in five major sequence clusters with strong bootstrap support, corresponding to TcI, TcII, TcIV, TcV and TcVI parasite DTUs. Reference strains for each DTUs are indicated by arrows (TcI: Raccoon70-EF576837, TcII: Tu18-AY367125, TcIV: 92122102-AY367124, TcV: SC43-AY367127, TcVI: CL-U57984 and VSC-. Numbers on tree branches indicate bootstrap support for each DTU clades. (**B**) Analysis of TcII, TcV and TcVI sequences confirmed that dog sequences clustered with reference sequences from each of these DTUs, and significant substructuring with strong bootstrap support was observed within TcII and TcVI DTUs (Only boostrap values above 50% are shown).

sequence clustered with TcV, and these DTUs were detected in US dogs for the first time (Fig 1B). Analysis of TcI sequences also revealed a marked substructuring with strong bootstrap support (S2 Fig). Most sequences clustered with TcIa and a few clustered with TcId.

These data confirmed previous observations in mammalian hosts and vectors from Louisiana and Texas [20–22,27]. Nonetheless, within DTUs, sequences derived from Louisiana dogs also differed significantly from those of reference strains from South America, suggesting potential sequence differentiation between north and south America for most DTUs as proposed for TcIV.[28]

Analysis of the geographic distribution of parasite DTUs from dogs indicated that TcI was found in dogs from all 11 shelters, while TcIV was detected in dogs from five shelters, and TcII, TcV and TcVI were found in dogs from two shelters (S1 Fig). However, these differences in distribution were likely due to a low sample size for several shelters.

## *T. cruzi* parasite diversity among vector, dogs and other mammalian hosts in Louisiana

We then compared *T. cruzi* diversity observed in dogs with that of parasites in *Triatoma sanguisuga* vectors, as well as in rodents and non-human primates from Louisiana, to get insights on how the different strains and DTU may be circulating among these hosts and vectors. Principal component analysis confirmed that highly similar strains from DTUs TcI, TcIV and the group of TcII, TcV and TcVI circulated among these hosts and vector species (Fig 2A), except for a single TcIV sequence reported in a non-human primate that diverged from the TcIV sequences from dogs, rodents and vectors. A separate analysis of the TcII, TcV, TcVI group further stressed that highly similar strains from each of these DTUs are circulating among these hosts and vectors (Fig 2B). Again, some notable exceptions were a few TcVI sequences from mice and rats that differed from TcVI sequences from the other hosts and vectors. Finally, analysis of the TcI DTU indicated a strong similarity of strains circulating in dogs and vectors (PERMANOVA, P = 0.28), while strains circulating in rodents and NHP represented a smaller subset of these strains (Fig 2C). This latter observation may be due to a sampling bias, as rodent and NHP sequences were derived from a limited number of animals and may not represent the full diversity of parasites circulating in these hosts. Haplotype networks including

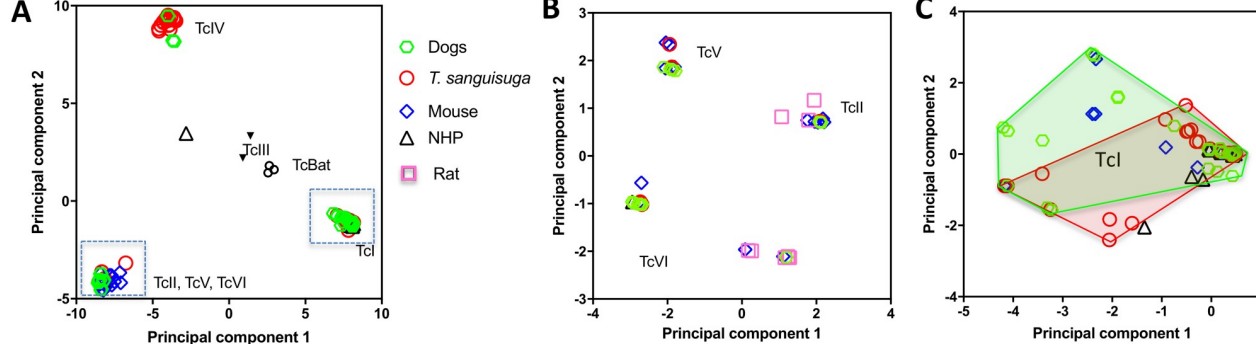

**Fig 2. Principal component analysis of *T. cruzi* mini-exon sequences from mammalian hosts and triatomines from Louisiana.** (**A**) Combined analysis of sequences from all DTUs, showing a strong clustering of sequences according to parasite DTUs. Points correspond to individual mini-exon sequences from *Triatoma sanguisuga*, dogs, mice, rats, and non-human primates (NHP)(color/symbol coded as indicated). Gray boxes indicate sequences from TcII/TcV/TcVI DTUs that are analyzed separately in (B) and TcI sequences analyzed in (C). (**B**) Analysis of TcII, TcV and TcVI sequences confirmed the clustering of sequences for each DTU. (**C**) Analysis of TcI sequences indicated some sequence diversity within this DTU, but sequences from *T. sanguisuga* and dogs did not present statistically significant differences (P = 0.28). Convex hull outlines for dogs and triatomines are shown.

sequences from vectors and multiple host species also indicated that many haplotypes from dogs were shared or closely related to the haplotypes from other species (S3 Fig), strengthening the idea that they are involved in common parasite transmission cycles in southern Louisiana.

Next, we analyzed the relative abundance of parasite sequence haplotypes within individual dog hosts. While a few animals presented a single sequence haplotype (5/39, 13%), the majority had multiple haplotypes (34/39, 87%), sometimes covering more than one DTU (13/39, 33%)(Fig 3 and S4 Fig). Nonetheless, a predominant haplotype was detected in most dogs and the average Berger-Parker dominance index was 0.57 ± 0.04. Ten dogs had mixed infections with TcI and TcIV (10/39, 26%), two had mixed infections with TcI, TcII, TcV and TcVI (5%), and two had mixed infections with TcI, TcII, TcIV, TcV and TcVI (5%). Overall, TcI parasites largely predominated in dogs, followed by TcIV, TcII, TcVI and TcV (Fig 4A). Comparison of DTU frequencies between dogs and *T. sanguisuga* vectors indicated that these varied significantly ($X^2$ = 40.15, P<0.0001). Indeed, while TcI predominated in both dogs and vectors, TcIV and TcV were detected more frequently in vectors, while TcII and TcVI were detected more frequently in dogs. Principal component analysis of DTU composition in dogs and vectors also indicated a significant difference in DTU composition between these two populations (PERMANOVA, P = 0.0005, Fig 4B). Thus, although similar parasite strains are found in both dogs and vectors in the region, their proportions may vary between vectors and hosts.

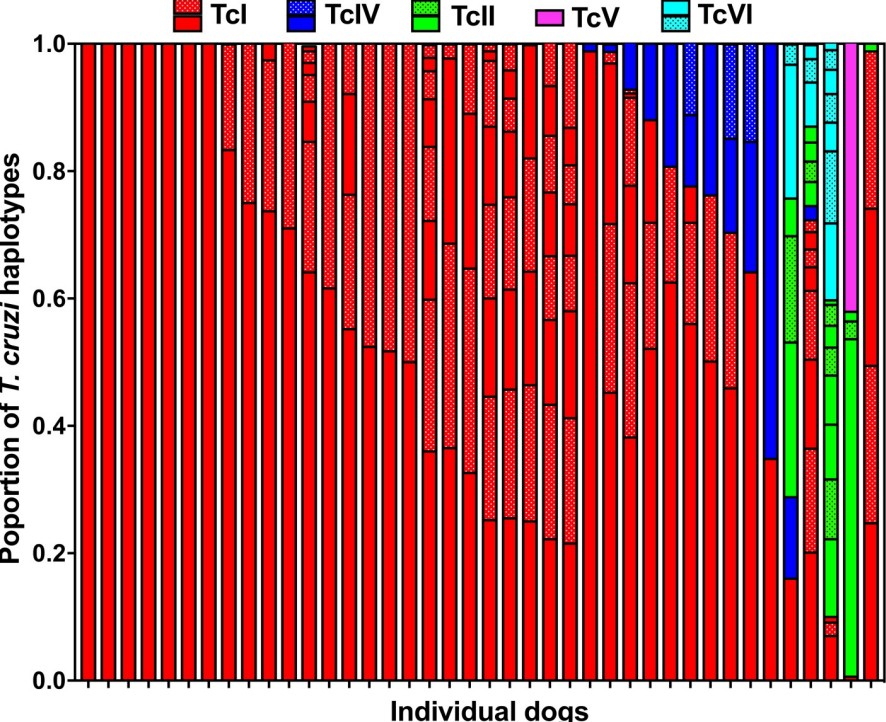

**Fig 3. Mini-exon sequence diversity in individual dogs.** Each bar represents an individual dog, and the proportion of sequences from each DTU as well as that of single sequence haplotypes within DTUs are color coded as indicated at the top of the graph. Twenty five dogs (64%) harbored only TcI parasites, 10 dogs had mixed infections with TcI and TcIV (26%), two had mixed infections with TcI, TcII, TcV and TcVI (5%), two had mixed infections with TcI, TcII, TcIV, TcV and TcVI (5%), and one had infection with TcI and TcII (2.5%).

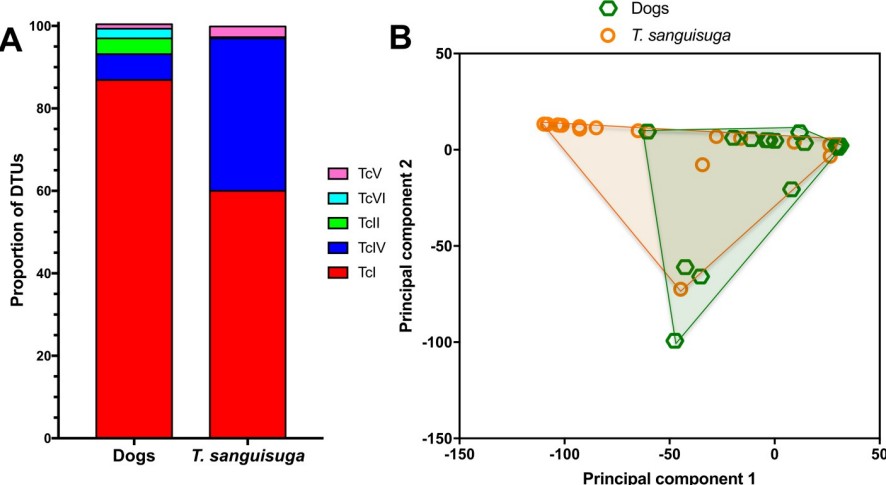

**Fig 4. Proportion of parasite DTUs in dogs and triatomines.** (**A**) Proportion of parasite DTUs is shown for shelter dogs and *Triatoma sanguisuga* from Louisiana. There was a significant difference in DTU proportions ($X^2 = 40.15$, $P<0.0001$). (**B**) Principal component analysis of DTU composition from dogs and *T. sanguisuga*. Points correspond to DTU composition in individual dogs and vectors, and convex hull outlines for each population are shown. PERMANOVA indicated a significant difference between the two populations ($P = 0.0005$).

## *T. cruzi* parasite diversity and serological diagnostic performance

Because a large proportion of PCR-positive dogs from our cohort had discordant serological tests or even were seronegative with these tests (see Methods, [9]), we tested the hypothesis that the poor performance of serological tests was due to parasite genetic diversity. Cases of infections with multiple DTUs are difficult to interpret due to the mixtures of genotypes, but among ten dogs with mixed infection with TcI and TcIV, five were seronegatives and three had discordant serology, and two were confirmed seropositives. Similar results were observed among dogs with co-infections including TcII, TcV and TcVI parasites.

We then focused our analysis on dogs with only TcI infections. Network analysis of parasite sequences indicated that different haplotypes were associated with discordant, negative and confirmed seropositivity, respectively (Fig 5A). Infections with both TcIa and TcId subgroups included discordant and seronegative dogs. We then compared the genetic distance of each sequence with that of the mini-exon from the WB1 strain which was used for the in-house serological diagnostic tests. However, sequences from discordant or seronegative dogs did not have a significantly greater genetic distance from the WB1 strain compared to sequences from confirmed seropositive dogs (ANOVA, P = 0.92). Thus, genetic differentiation based on the mini-exon marker did not strictly reflect the antigenic differentiation of the strains.

On the other hand, analysis of mini-exon sequence haplotypes by principal component followed by PERMANOVA indicated that dogs with negative, discordant or confirmed positive *T. cruzi* serology harbored TcI parasites with statistically different mini-exon sequences (P = 0.004, Fig 5B). Parasite sequences from ELISA reactive dogs even differed from those of western blot reactive dogs, while those from confirmed positive dogs partly overlapped the sequences from these discordant samples.

Taken together, these data strongly supported the hypothesis that parasite genetic diversity is a key factor affecting serological test performance, which needs to be taken into account for the development of improved tests.

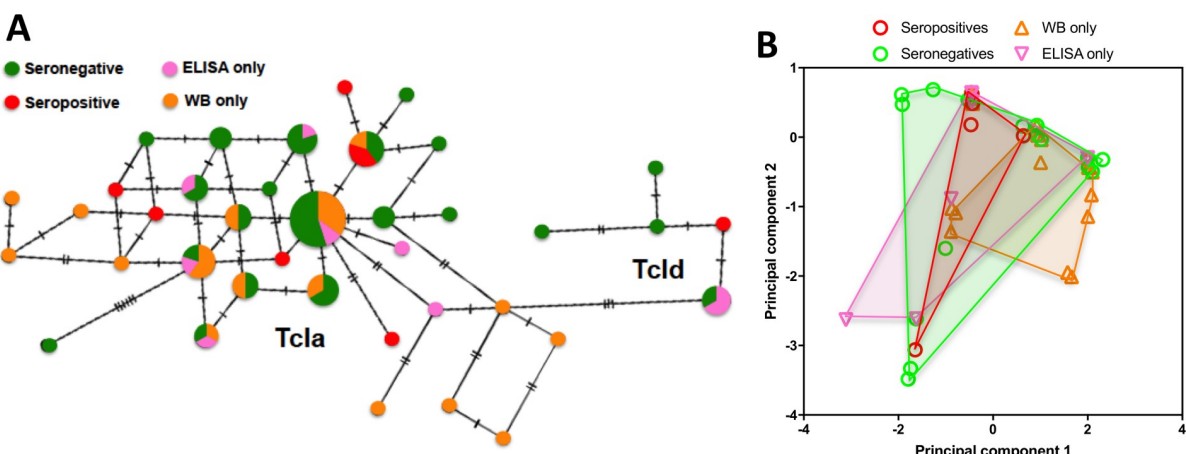

**Fig 5. Mini-exon sequence diversity and serological testing of dogs.** (**A**) TCS network of TcI mini-exon sequences from dogs with different serological status, including confirmed seropositive (at least two reactive tests), seronegative, and discordant with only a reactive ELISA or western blot, respectively. The size of nodes is proportional to the number of sequences, and ticks on branches indicate the number of mutations between sequences. Sequences clustered with TcIa and TcId references. (**B**) Principal component analysis of TcI mini-exon sequences from dogs with the indicated serological status. Points indicate individual sequences, and convex hulls for the respective groups are shown. Principal component 1 and 2 represented 28.2% and 18.5% of the variation, respectively. PERMANOVA indicated a statistically significant difference among groups, *P* = 0.0004.

## Discussion

The characterization of *T. cruzi* strains circulating in vertebrate hosts and vectors is key to understanding parasite transmission cycles, and assessing the potential associations of parasite strains with epidemiologic characteristics such as host specificity or pathogenesis profiles, and risk for human infection. In the US, limited information is available on these aspects. Initial observations described only TcI and TcIV in mammalian hosts in the southern US [18,29,30], but recent work indicated the circulation of additional DTUs including TcII, TcV and TcVI in some vector and host species, including humans [20–22,27,31]. We analyzed here *T. cruzi* diversity in one of the largest cohort of infected shelter dogs, to assess parasite transmission cycles in southern Louisiana.

Our analysis confirmed the extensive diversity of strains infecting shelter dogs, covering TcI and TcIV as before, and confirming the presence of TcII, TcV and TcVI for the first time in dogs. In addition, we found that TcI included TcIa and TcId, as observed in rodents [21] and vectors [27] in the region. Our phylogenetic analysis also revealed some significant local differentiation of strains within each DTU for which multiple sequences were available. Thus, *T. cruzi* strains from the southern US may differ from those from Central or South America within each DTU. Together, these results point to a high diversity of *T. cruzi* strains in the southern US and warrant additional studies in other regions for an accurate description of strains circulating in the country and how they may be geographically structured. Indeed, previous studies failed to detect major geographic structuring of TcI strains across the Americas [32], possibly because a smaller sequence fragment of the mini-exon marker was used. Our results further highlight that deep sequencing for genotyping has an increased sensitivity to detect infections with *T. cruzi* parasites from multiple DTUs, which are often lost when genotyping by PCR only or even using Sanger sequencing of PCR products.

Infections with multiple parasite strains also appeared common in dogs, although the exact strain number is difficult to estimate based on the mini-exon marker. Indeed, as discussed before, it is a multicopy marker with paralogous variations [23,33], but haplotype frequencies and multiple DTUs do support the idea of frequent infections with several parasite strains/

DTUs in dogs. Indeed, TcVI strains TCC and CL Brener only present a few percent of their paralogous mini-exon sequences that clustered with TcII DTU [23]. Thus, the observation of TcVI and TcII sequences in comparable and high proportions in dogs is more compatible with the presence of parasite strains from multiple DTUs. The implication of such infections for clinical disease progression are largely unknown, but studies in mouse models suggest potential interactions among parasite strains that can affect disease progression [34–36]. The relatively high predominance of a sequence haplotype in each dog, as indicated by the Berger index, suggests that although multiple infections are present, specific genotypes may be preferentially amplified, while the host may better control others.

In that respect, comparison of parasite haplotypes found in shelter dogs with those from other mammalian hosts and vectors from Louisiana indicated that highly similar haplotypes are in fact circulating among these hosts and vectors in the region. Thus, these vectors and hosts are involved in a shared *T. cruzi* parasite transmission cycle. However, we also detected significant differences in the frequency of DTUs between vectors and shelter dogs. Indeed, TcIV and TcV were more frequent in vectors, while TcII and TcVI were more frequent in dogs. These data are more difficult to interpret as these differences may reflect some sampling bias, a geographic or temporal structuration, or different parasite selection/dominance patterns in each species. Although large, our sample is still limited to assess the potential spatial structure of *T. cruzi* populations. Differences in parasite development in triatomines and dogs due to competition for resources and immune selection pressure is an hypothesis worth testing A previous study suggested that ecological host-fitting rather than co-evolution is the dominant mechanism involved in the diversification of *T. cruzi*-host associations [37]. This mechanism implies that phenotypic plasticity of parasites allows them to be pre-adapted to new resources/hosts. Expanding these observations to larger samples from additional locations and host species would allow to better assess how *T. cruzi* strains and DTU may partition among species and be involved in partially overlapping transmission cycles.

One of the most immediate implications of the circulation of genetically diverse *T. cruzi* strains in hosts is for serological diagnostics. Indeed, antigens used in diagnostic tests need to closely match those from strains actually circulating in the population to ensure the reliable identification of infection, and strong antigenic variation will cause test performance to be low. As noted above, discrepancies among serological tests in North America can be high in dogs [9,11] as well as in humans [12–16], raising concerns for the surveillance of *T. cruzi* infection. Also, our phylogenetic analysis of the mini-exon marker suggests some genetic differentiation within DTUs based on the geographic origin of the strains. Analysis of infections with multiple DTUs are difficult to interpret, but it is noticeable that some dogs with co-infections with TcI and TcIV or with TcI, TcII and TcVI were seronegative with three tests, two of which were based on a local TcI strain. In humans, many patients with a discordant or absent serological response do have detectable T cell responses against *T. cruzi* [38,39], but these are more difficult to assess. Detailed analysis of TcI sequences revealed that different haplotypes were associated with discordant, negative and confirmed seropositivity, respectively, but the genetic distance of each sequence to that of the local strain used for diagnostic was not linearly associated with the serological test outcome. Nonetheless, dogs with negative, discordant or confirmed positive *T. cruzi* serology harbored TcI parasites with statistically different mini-exon sequences, which strongly supports the hypothesis that parasite genetic diversity is a key factor affecting serological test performance. These observations are in agreement with a previous study in which specific mini-exon sequence haplotypes could be statistically associated with inconclusive/negative ELISA test results in human samples [40]. However, minimal differences in mini-exon sequences may translate into larger antigenic variation, as evolutionary forces acting on the mini-exon likely differ from those acting on parasite antigens. Together,

these results suggest that the identification of highly conserved antigens may greatly improve *T. cruzi* serological diagnostic performance.

In conclusion, we report here that shelter dogs in southern Louisiana are infected with a wide diversity of *T. cruzi* strains, including TcI and TcIV and for the first time TcII, TcV, and TcVI DTUs. Infections with multiple parasite strains/DTUs were also frequent, which may have implications for disease progression. Furthermore, while similar strains circulate in mammalian hosts and vectors in the region, TcIV and TcV were more frequent in vectors, while TcII and TcVI were more frequent in dogs, suggesting different selection pressures leading to different strain dominance patterns. These observations suggest that ecological host-fitting rather than co-evolution is the dominant mechanism involved in the diversification of *T. cruzi*-host associations. Finally, the observation that dogs with negative, discordant, or confirmed positive *T. cruzi* serology harbored TcI parasites with different mini-exon sequences strongly supports the hypothesis that parasite genetic diversity is a key factor affecting serological test performance. Thus, the identification of conserved parasite antigens should be a high priority for the improvement of current serological tests.

## Supporting information

**S1 Table. List of mini-exon sequences from dogs.**
(PDF)

**S2 Table. List of mini-exon sequences from vectors and other mammalian hosts.**
(PDF)

**S1 Fig. Geographic distribution of animal shelters and genotyped *T. cruzi* parasite populations from dogs in Louisiana.** Pies charts for each shelter indicate the distribution of *T. cruzi* DTUs, with their size proportional to the number of dogs analyzed at each shelter.
(TIF)

**S2 Fig. Phylogenetic analysis of TcI and TcIV mini exon sequences derived from shelter dogs.** Maximum likelihood analysis of TcI (**A**) and TcIV sequences (**B**) is shown. Reference strains are indicated by arrows, and numbers on branches indicate bootstrap support for values over 50%.
(TIF)

**S3 Fig. Mini-exon haplotype networks from mammalian hosts and triatomine vectors from southern Louisiana.** TCS networks were constructed based on mini-exon sequences from dogs, mice, rats, non-human primates (NHP) and *Triatoma sanguisuga* from southern Louisiana. Nodes represent haplotypes, with their size proportional to the number of sequences, and they are color-coded by species. Ticks on branches indicate the number of mutations from one haplotype to the next. (**A**) TcI DTU. Arrows point to reference sequences from strains Raccoon70 (TcIa) and SylvioX10 (TcId). (**B**) TcII, TcV and TcVI DTUs. Arrows point to reference sequences from strains Tu18 (TcII), SC43 (TcV), and CL (TcVI).
(PDF)

**S4 Fig. Examples of mini-exon haplotype networks from individual dogs.** TCS networks were constructed based on mini-exon sequences from individual dogs (**A, B, C, D, E**). Nodes represent haplotypes, with their size proportional to their proportion as indicated, and they are color-coded by DTUs. Ticks on branches indicate the number of mutations from one haplotype to the next.
(PDF)

## Author Contributions

**Conceptualization:** Eric Dumonteil, Claudia Herrera.

**Formal analysis:** Eric Dumonteil, Preston Marx, Garry Balsamo, Claudia Herrera.

**Funding acquisition:** Eric Dumonteil, Preston Marx.

**Methodology:** Ardem Elmayan, Alicia Majeau, Weihong Tu, Brandy Duhon, Wendy Wolfson.

**Project administration:** Eric Dumonteil.

**Writing – original draft:** Eric Dumonteil.

**Writing – review & editing:** Ardem Elmayan, Alicia Majeau, Weihong Tu, Brandy Duhon, Preston Marx, Wendy Wolfson, Garry Balsamo, Claudia Herrera.

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
