## [Decision Letter · Decision Letter 0]

6 Jul 2020

Dear Dr. Dumonteil,

Thank you very much for submitting your manuscript "Genetic diversity of Trypanosoma cruzi parasites infecting dogs in southern Louisiana sheds light on parasite transmission cycles and serological diagnostic performance" for consideration at PLOS Neglected Tropical Diseases. As with all papers reviewed by the journal, your manuscript was reviewed by members of the editorial board and by several independent reviewers. In light of the reviews (below this email), we would like to invite the resubmission of a significantly-revised version that takes into account the reviewers' comments. 

Overall, this is, potentially, an interesting contribution. However, there are important comments from the reviewers that the authors should address, particularly regarding the vector data.

We cannot make any decision about publication until we have seen the revised manuscript and your response to the reviewers' comments. Your revised manuscript is also likely to be sent to reviewers for further evaluation.

Sincerely,

Ananias A. Escalante, PhD

Associate Editor

Marcelo Ferreira

Deputy Editor

Overall, this is, potentially, an interesting contribution. However, there are important comments from the reviewers that the authors should address, particularly regarding the vector data.

Reviewer's Responses to Questions

**Key Review Criteria Required for Acceptance?**

**Methods**

-Are the objectives of the study clearly articulated with a clear testable hypothesis stated?

-Is the study design appropriate to address the stated objectives?

-Is the population clearly described and appropriate for the hypothesis being tested?

-Is the sample size sufficient to ensure adequate power to address the hypothesis being tested?

-Were correct statistical analysis used to support conclusions?

-Are there concerns about ethical or regulatory requirements being met?

Reviewer #1: (No Response)

Reviewer #2: All method-related aspects have been covered according to current good practice.

Reviewer #3: The mini-exon gene is multicopy and it is thus an odd choice for this type of work as you are dealing with multiple variable copies within a single genome as well as variation among strains in mixed infections. It would have made much more sense to use a single-copy locus that provides enough DTU resolution. In that context the description of how sequences are mapped and assembled (?) for each sample has to be expanded (p. 7). Checking the cited reference #19 there is really not enough information there to know what was done and how. No information on quality control checks, removal or chimeras and identification of paralogous copies.

**Results**

-Does the analysis presented match the analysis plan?

-Are the results clearly and completely presented?

-Are the figures (Tables, Images) of sufficient quality for clarity?

Reviewer #1: I know that the sequence data has been deposited in GenBank, but it would be helpful if the authors would include a supplementary table that lists the accession numbers along with the identity and location of the 

dogs from which these were obtained. Otherwise, it may be difficult or impossible for a reader to determine

when two or more sequences were obtained from the same host individual.

Reviewer #2: Figures and other results are clearly presented, no flaw detected.

Reviewer #3: Please provide examples of diversity within a single sample to get a sense of intragenomic diversity in the locus. You can do this with in a Supplementary figure or table.

Why is the TcIII reference not shown in Figure 1? This is odd given that it is well known that TcV and TcVI are hybrids between TcII and TcIII. Are mini-exon sequences from TcV and TcVI completely different from those of TcII and TcIII or do they have a mix of haplotypes from each type as observed in any other nuclear loci? Given the hybrid nature of TcV and TcVI it is odd that you did not pick up expected TcII sequences in those samples (but in Figure 4 every case with TcV or VI shows a TcII, see comment below). How do you explain this finding?

Figure 2: it is not clear where the data from vectors comes from, what is the sample size, etc. There was no information in the manuscript about those samples. It is also not obvious why presenting the data in a PCA plot is better than showing again a display in a phylogenetic tree to depict more clearly the relationships between samples collected in the dogs and the vectors (from unknown precedence). Are there that many non-dog samples that showing them in a tree with select dog samples is not possible? There is so much overlap among the similar samples in the PCA that it is not possible to make sense of the meaning of the findings. For instance: the TcI cluster in part A seems to be dominated by Dog samples, but there are other samples covered by the green dog circles, what are they?

Figure 4: this is a cool figure and one that directly shows the value of next gen sequencing to determine mixed infection levels. However, why are there only 40 samples shown? In the methods you said that 73 dogs were sampled. Something interesting is that every case with TcV or TcVI has TcII. The authors interpret this as evidence of strain coocurrence but it may be just a case of the two haplotypes from hybrids being picked up in the samples.

**Conclusions**

-Are the conclusions supported by the data presented?

-Are the limitations of analysis clearly described?

-Do the authors discuss how these data can be helpful to advance our understanding of the topic under study?

-Is public health relevance addressed?

Reviewer #1: (No Response)

Reviewer #2: Results are fully discussed and conclusions supported by the data. Last paragraph of Discussion is just a repetition of said concepts and should be deleted. The study has some implications for better diagnosis of canine infections.

A major result needs more attention: Can the authors discuss potential limitations of their approach in terms of unspecific results, for example, the finding of TcV and TcVI? Did any other authors in the US (other than this research team) reported the finding of these DTUs there? Were their methods limited in this respect? Any suggestion on the source of these DTUs (e.g., humans?

Reviewer #3: A major conclusion is the finding of a connection between genetic diversity in TcI infections and performace of serological tests, which seems very interesting and important. I just wonder how relevant is it given the small sample sizes. In Fig 5B the seronegatives cover most of the PCA plot and have complete overlap with the seropositives.

**Editorial and Data Presentation Modifications?**

Reviewer #1: There are several sentences containing minor grammatical errors or confusing wording. Suggested corrections are listed below:

line 62: differential selection of lineages.

line 163: was assessed

line 176: was assessed

line 182: In total, 73 samples ... were

line 265: Overall, TcI parasites 

line 327: ticks on branches

line 354: point to a high diversity

line 363: implication of such infections

line 368: while the host may better control others.

line 382: and DTUs may partition among host species

line 385: for serological diagnostics.

line 394: two of which were based on a

line 397: the genetic distance of each sequence to that of the local

Reviewer #2: check a few typos, e.g. line 363

Reviewer #3: Sentence in line 78: provide reference

**Summary and General Comments**

Reviewer #1: This manuscript describes genetic variation at a nuclear barcoding marker in Trypanosoma cruzi parasites 

infecting domestic dogs in animal shelters in southern Louisiana. To resolve mixed infections, the 

investigators used PCR amplification of the mini-exon marker followed by deep sequencing on a MiSeq platform. This approach yielded genetic data for 39 out of 73 PCR-positive animals. Analysis of this data 

revealed the following points:

1) Mixed infections (as evidenced by the mini-exon locus) are common in dogs in this region.

2) Genetic variation in canine infections is high in this region, with the authors detecting several sequence

types (DTU's) for the first time within the US.

3) North American isolates are genetically distinct from South American reference strains.

4) The distribution of mini-exon genotypes differs between the domestic dogs sampled in this study and 

Triatomine vectors investigated in an earlier study.

5) Variation in the mini-exon sequence is significantly correlated with serological test performance, 

with certain mini-exon sequences being associated with negative or discordant test results more frequently 

than expected by chance. 

On the whole, this is an solidly-designed study which makes an interesting contribution to our 

understanding of canine trypanosomiasis in the southeastern US. However, I have one major criticism, pertinent to result #4. The authors perform a chi-squared test comparing the frequencies of mini-exon haplotypes isolated from dogs with those isolated in vectors and find that these distributions are significantly different. They then conclude that "these data strongly suggest the occurrence of strain/DTU differential selection in vectors and hosts". While this conclusion may be correct, I believe that it is unwarranted by the present study for two reasons.

One difficulty is with the validity of the chi-squared test for frequency data sampled in structured populations. A core assumption of this test is that the samples are independent of one another. However, 

in the present study, multiple dogs have been sampled from the same shelter and, by virtue of their geographical proximity, arguably are more likely to share related parasite genotypes than dogs sampled from distant shelters. Although the authors do not explicitly test for geographical structure in their data, Figure S1 demonstrates that there is geographical variation in the haplotype distribution within shelter dogs. Likewise, because most infections are mixed, many of the samples used to calculate the haplotype

frequencies will have been collected from the same host. However, this too may lead to violations of the

independence assumption if either drift or selection is occurring within individual hosts. For these reasons, 

I am skeptical of the significance of this test result. As an aside, it may be of interest to explicitly test

for geographical structure in this data, noting that there are two levels of organization: sequences within

hosts and hosts within structures, e.g., use a hierarchical AMOVA.

A potentially more serious objection comes from the fact that we are not told anything about the 

study design used to generate the vector data, which is from an unpublished manuscript. Unless the 

hosts and the vectors were sampled at the same time and in the same locations using a balanced sample design, we cannot exclude the possibility that differences between the samples are due to confounding

spatial or temporal variation in the genetic composition of the parasite population. This effect will be

exacerbated if there are clonal expansions or non-random interactions between the hosts and vectors.

I believe that the authors should take two steps to address these concerns. First, they should be much 

more cautious when interpreting these results and note both the statistical and interpretive challenges

that the data presents. Secondly, it would be helpful for them to include some information concerning

when and where the vectors were sampled.

Reviewer #2: Great work, very well written and presented, some very surprising results (at least for me), further studies for confirmation needed.

Reviewer #3: See comments above for all suggestions

PLOS authors have the option to publish the peer review history of their article (what does this mean?). If published, this will include your full peer review and any attached files.

Reviewer #1: Yes: Jay Taylor

Reviewer #2: No

Reviewer #3: No
---

## [Decision Letter · Decision Letter 1]

15 Sep 2020

Dear Dr. Dumonteil,

Thank you very much for submitting your manuscript "Genetic diversity of Trypanosoma cruzi parasites infecting dogs in southern Louisiana sheds light on parasite transmission cycles and serological diagnostic performance" for consideration at PLOS Neglected Tropical Diseases. As with all papers reviewed by the journal, your manuscript was reviewed by members of the editorial board and by several independent reviewers. The reviewers appreciated the attention to an important topic. Based on the reviews, we are likely to accept this manuscript for publication, providing that you modify the manuscript according to the review recommendations. 

Sincerely,

Ananias A. Escalante, PhD

Associate Editor

Marcelo Ferreira

Deputy Editor

Reviewer's Responses to Questions

**Key Review Criteria Required for Acceptance?**

**Methods**

-Are the objectives of the study clearly articulated with a clear testable hypothesis stated?

-Is the study design appropriate to address the stated objectives?

-Is the population clearly described and appropriate for the hypothesis being tested?

-Is the sample size sufficient to ensure adequate power to address the hypothesis being tested?

-Were correct statistical analysis used to support conclusions?

-Are there concerns about ethical or regulatory requirements being met?

Reviewer #1: (No Response)

Reviewer #2: I found the responses to other reviewers' queries appropriate. 

Since one of the conclusions of this paper is related to the mismatch between serodiagnosis and parasite genetic makeup, the authors should provide more details on serodiagnostic methods, cut-off values, sensitivity and specificity for each of them, and not just provide the reference for a previous work. More in Results below.

PERMANOVA: software used?

Reviewer #3: The authors have improved the description of methods and they are now satisfactory.

**Results**

-Does the analysis presented match the analysis plan?

-Are the results clearly and completely presented?

-Are the figures (Tables, Images) of sufficient quality for clarity?

Reviewer #1: (No Response)

Reviewer #2: I found the responses to other reviewers' queries appropriate. 

In reference to the association between serodiagnosis outcomes and parasite genetic makeup, the authors should provide a contingency table that summarizes these specific results. As one of the claims of this manuscript is that serodiagnosis should be improved, these relationships should be made explicit. (In this regard, consider adding a reference to similar criticisms put forward by Tarleton and Cooley, this subject goes way back and beyond canine serodiagnosis).

Reviewer #3: Thanks for adding Figure S4 to help provide a sense of intragenomic diversity in the miniexon locus in 5 individual dogs with different infection profiles. In the figure legend please remove the end of the first sentence (“in each panel”) because it is confusing.

Thanks for the clarification about what the presence of multiple haplotypes means and why TcIII is not present in infections from hybrid DTUs TcV and TcVI. The lack of resolution to determine the exact nature of mixed infections using miniexon sequences underscore the importance of also using single copy genes that will provide better resolution even if the authors claim that low parasitemia reduces chance of positive PCRs. Designing primers for PCR products of similar size for multiple markers that can be combined for amplicon next gen sequencing should provide better resolution. Food for thought for future work.

The clarification about PCA and added figures are very useful.

**Conclusions**

-Are the conclusions supported by the data presented?

-Are the limitations of analysis clearly described?

-Do the authors discuss how these data can be helpful to advance our understanding of the topic under study?

-Is public health relevance addressed?

Reviewer #1: (No Response)

Reviewer #2: I found the responses to other reviewers' queries appropriate, and limitations clearly described

Reviewer #3: Generally the work is well presented, conclusions well supported, and public health relevance is pretty obvious. Something to have in mind when trying to interpret negative serological tests in individuals that are infected with the parasite (based on a positive PCR) is that serological tests only detect one aspect of the immune response. T cell responses are as important or more than antibody production but they are much harder to measure. It is possible that most of the immune response against the parasites or against some of the DTUs is driven by T cells. Some mention of this in the discussion or conclusions would be useful.

**Editorial and Data Presentation Modifications?**

Reviewer #1: (No Response)

Reviewer #2: No suggestion on these items.

Reviewer #3: (No Response)

**Summary and General Comments**

Reviewer #1: The authors have adequately addressed the concerns raised in my initial review of their manuscript. Apart from a few minor changes in wording (suggested below), I think that the revised manuscript is suitable for publication.

Suggested minor corrections:

p. 41: However, TcIV and TcV were sampled more frequently in vectors, while TcII and TcVI were sampled more frequently in dogs.

p. 188: The list of sequences ...

p. 307: ... were detected more frequently in vectors, while ...

p. 382: ... key to understanding parasite transmission cycles, and assessing the ...

p. 446: Expanding these observations to larger samples from additional locations and host species would ...

p. 463: ... to that of the local strain used for diagnostics ...

Reviewer #2: Interesting paper. It may stimulate others to reevaluate the occurrence of parasite DTUs in the USA.

Reviewer #3: See comments above and my first review

PLOS authors have the option to publish the peer review history of their article (what does this mean?). If published, this will include your full peer review and any attached files.

Reviewer #1: Yes: Jay Taylor

Reviewer #2: No

Reviewer #3: No
---

## [Decision Letter · Decision Letter 2]

29 Oct 2020

Dear Dr. Dumonteil,

We are pleased to inform you that your manuscript 'Genetic diversity of Trypanosoma cruzi parasites infecting dogs in southern Louisiana sheds light on parasite transmission cycles and serological diagnostic performance' has been provisionally accepted for publication in PLOS Neglected Tropical Diseases.

Best regards,

Ananias A. Escalante, PhD

Associate Editor

Marcelo Ferreira

Deputy Editor

Reviewer's Responses to Questions

**Key Review Criteria Required for Acceptance?**

**Methods**

-Are the objectives of the study clearly articulated with a clear testable hypothesis stated?

-Is the study design appropriate to address the stated objectives?

-Is the population clearly described and appropriate for the hypothesis being tested?

-Is the sample size sufficient to ensure adequate power to address the hypothesis being tested?

-Were correct statistical analysis used to support conclusions?

-Are there concerns about ethical or regulatory requirements being met?

Reviewer #1: (No Response)

Reviewer #2: I have addressed these questions in my previous review. The authors' revisions in response to the reviewers' comments are satisfactory.

Reviewer #3: (No Response)

**Results**

-Does the analysis presented match the analysis plan?

-Are the results clearly and completely presented?

-Are the figures (Tables, Images) of sufficient quality for clarity?

Reviewer #1: (No Response)

Reviewer #2: I have addressed these questions in my previous review. The authors' revisions in response to the reviewers' comments are satisfactory.

Reviewer #3: (No Response)

**Conclusions**

-Are the conclusions supported by the data presented?

-Are the limitations of analysis clearly described?

-Do the authors discuss how these data can be helpful to advance our understanding of the topic under study?

-Is public health relevance addressed?

Reviewer #1: (No Response)

Reviewer #2: I have addressed these questions in my previous review. The authors' revisions in response to the reviewers' comments are satisfactory.

Reviewer #3: (No Response)

**Editorial and Data Presentation Modifications?**

Reviewer #1: (No Response)

Reviewer #2: (No Response)

Reviewer #3: (No Response)

**Summary and General Comments**

Reviewer #1: This is an interesting and well-written paper and I have no additional concerns at this stage.

Reviewer #2: I have addressed these questions in my previous review. The authors' revisions in response to the reviewers' comments are satisfactory.

Reviewer #3: The revision is satisfactory. I have no concerns about the methodology or conclusions in the revision. There is still a lingering question about the Permanova test showing significant differences between T cruzi I variants associated with seronegative or seropositive dogs which seems hard to believe given the level of overlap shown in the network and PCA plots (Figure 5), but I have to trust the authors have run the tests carefully.

PLOS authors have the option to publish the peer review history of their article (what does this mean?). If published, this will include your full peer review and any attached files.

Reviewer #1: **Yes: **Jay Taylor

Reviewer #2: No

Reviewer #3: No

---

## [Editor Report · Acceptance letter]

9 Dec 2020

Dear Dr. Dumonteil,

We are delighted to inform you that your manuscript, "Genetic diversity of *Trypanosoma cruzi* parasites infecting dogs in southern Louisiana sheds light on parasite transmission cycles and serological diagnostic performance," has been formally accepted for publication in PLOS Neglected Tropical Diseases.

Best regards,

Shaden Kamhawi

co-Editor-in-Chief

Paul Brindley

co-Editor-in-Chief
